# Identifying Meaningful Vulnerability Report in Common Weakness Enumeration

*Abstract*—The vulnerabilities in Open Source Software (OSS) code, particularly critical ones, offer attackers numerous opportunities, leading to significant economic losses for users. This has driven the development of various models to identify these vulnerabilities. However, previous models often used shallow neural networks with a single feature extraction method, failing to capture deep feature representations. To address this, we propose Vuln-Detector, an approach for automatically identifying dangerous Issue Reports (IRs). Vuln-Detector comprises three components: the Knowledge Bank component, which stores information about Common Weakness Enumeration (CWE) to enhance learning; the Matching component, which measures the similarity between a security report and CWE categories; and the Voting component, which determines whether a report is related to a code vulnerability. We validated our approach through experiments on 3,937 No Security Vulnerability Reports (NSVRS) from 1,390 OSS projects on GitHub. Vuln-Detector achieved a precision of 42%, recall of 73%, F1-score of 53%, AUROC of 98%, and AUPRC of 41%. Compared to the current state-of-the-art, it shows a relative improvement of 11% in precision, 5% in AUPRC, and 8% in F1-score. The results demonstrate that Vuln-Detector effectively identifies vulnerability-related IRs.

*Index Terms*—Vulnerability, Open Source Software Security, Deep Learning

## I. INTRODUCTION

Open source is widely used in the software industry as an efficient development model. Despite its numerous benefits, OSS openness brings many security threats [1], [2]. Attackers exploit publicly available vulnerability information to launch attacks, disrupting development and increasing security maintenance for teams. Traditional approaches follow a Coordinated Vulnerability Disclosure (CVD) process [3], where vulnerabilities are coordinated with vendors for patching before public disclosure. However, many organizations lack systematic security reviews and necessary testing standards [4]. Vulnerability reports, often submitted to public Issue Tracking Systems (ITS) before formal disclosure, create a window for attackers. This makes such reports, known as dangerous IRs, significant security risks [5].

Therefore, it is crucial to correctly label reported IRs as dangerous or not. However, manually checking these IRs is highly inefficient and time-consuming. Previous approaches [6] mostly train models based on the textual descriptions of IRs as input and output of the IR categories.

Peters et al. [5] introduced the FARSEC framework, which identifies and removes non-security error reports using security keywords, but did not consider the impact of security crosswords. Le et al. [7] proposed DeepCVA, achieving a higher MCC than other baselines by using an attention-based convolutional gated recurrent unit and context-aware features. Oyetoyan et al. [8] improved model generalization by using highly customized TF-IDF values as security keywords. Most previous approaches used datasets limited to specific projects, degrading performance on unseen projects. Challenges still exist in labeling IRs.

**Lack of deep features in the input IRs:** Vulnerability reports have high dimensionality and sparsity in their feature vectors, making feature extraction crucial for classification. Most previous models perform convolution, classification, or regression without post-processing semantic features, leading to lower accuracy. Shallow network models significantly weaken detailed features and fail to capture global contextual information, leading to the loss of key details. Long IR sequences, if not properly extracted, lose valuable information, affecting classification tasks.

To address the above challenges, we propose Vuln-Detector for accurate and efficient IR labeling. Vuln-Detector automatically identifies dangerous IRs early in vulnerability generation and consists of three components: an external storage component, a feature extraction component, and a classification component. The external storage component provides a vast repository of vulnerability information for model training, enhancing detection capabilities. The feature extraction component employs a self-attention mechanism to capture richer global contextual information and integrate feature information. The classification component maps the extracted feature vectors to the same feature space for classification.

Our contributions are as follows:

- We propose an automated approach that flattens the feature tensors of the IRS and external anchor obtained through BERT compression, performs multidimensional feature extraction and then fuses the extracted features for issue report classification. We also add the self-attention mechanism for feature repair and extraction, enhancing the approach's understanding of the overall content of sentences.

- We conduct experiments demonstrating the effectiveness of Vuln-Detector. Our approach excels in Precision, F1-score, AUROC, and AUPRC metrics. Precision surpasses the best baseline by 6%, F1-score by 5%, and AUPRC by 4%. These results indicate that Vuln-Detector effectively identifies dangerous vulnerability reports and outperforms state-of-the-art methods. The code and dataset for Vuln-Detector are open-source [9].

The rest of this paper is organized as follows. Section

II introduces the motivation behind our work. The main components of Vuln-Detector are described in Section III. Sections IV and V present the experimental setup and results, respectively. Threats to validity are discussed in Section VI. Related work is reviewed in Section VII. Finally, Section VIII provides a summary of our work.

## II. MOTIVATION

In this section, we will elucidate the application background of Vuln-Detector (Section A) and its key technologies (Section B) to introduce the motivation of this paper.

### A. Background

OSS is often developed by programmers, tested, and then used by end-users who provide feedback [10]. Vulnerability information is initially submitted as Incident Reports (IRs) to Issue Tracking Systems (ITS) before formal disclosure. However, vulnerabilities are sometimes disclosed long after their submission. Such delays can be risky, as IRs often contain detailed information about vulnerabilities that can be exploited by hackers, posing significant risks to OSS users and maintainers [11].

Data shows that 98.7% of Non-Secure Vulnerability Reports (NSVRs) are created before the corresponding National Vulnerability Database (NVD) disclosure dates, indicating that most IRs are at risk of leaking critical vulnerability information, leaving OSS users exposed to undetected security threats [12]. This situation reduces the time available for remediation.

According to Hazra et al. [13], large-scale software systems can harbor millions of bugs, making manual inspection of bug reports both tedious and error-prone. Consequently, developing a model to automatically identify dangerous IRs is crucial for enhancing vulnerability management and improving the efficiency of fixing vulnerabilities in open-source software.

### B. Key Techniques

The textual descriptions of NSVRs can vary widely across different Common Weakness Enumeration (CWE) categories. As shown in Figure 1, categories like "Improper Removal of Sensitive Information Before Storage or Transfer" and "ASP.NET Misconfiguration" exhibit notable differences in their descriptions. Previous methods [8] have extracted features from IRs or used TF-IDF values to train models, but these methods struggle to capture the full range of vulnerability knowledge and overall semantic information.

Jawahar et al. [14] noted that as BERT layers increase, surface-level information can become diluted, and the CLS representation may lose detailed sentence structure information due to its fixed-length nature. To address this, we use additional feature extraction and an attention mechanism to optimize local information and mitigate the loss of detail in the pooled sequence representation.

## III. FRAMEWORK OF THE PROPOSED APPROACH

In this section, we first provide an overview of our approach in Section A. Then, we proceed to describe the components of the model in detail in Sections B - D.

Improper Removal of Sensitive Information Before Storage or Transfer
Description:The product stores, transfers, or shares a resource that contains sensitive information, but it does not properly remove that information before the product makes the resource available to unauthorized actors.

ASP.NET Misconfiguration: Creating Debug Binary
Description:Debugging messages help attackers learn about the system and plan a form of attack

Fig. 1. Examples of CWE textual descriptions.

### A. Model Overview

The architecture of Vuln-Detector, depicted in Fig. 2, consists of three components: a) Knowledge Bank, b) Matching component, and c) Voting component. The Knowledge Bank provides extensive vulnerability information for learning descriptions and improving matching success. The Matching component uses anchors from the CWE tree structure and employs a Siamese network with BERT as the shared encoder for text embedding and feature extraction. These vectors are classified through a linear layer. The Voting component assigns scores to each output based on CWE matching, labeling the vulnerability type as NSVR if the score exceeds a threshold; otherwise, it is labeled as a Specific Vulnerability Report (SVR).

### B. Knowledge Bank component

Knowledge Bank is used to store information about different categories of CWE, providing a wealth of vulnerability knowledge to the model. CWE provides descriptions of weaknesses, and each weakness corresponds to vulnerabilities related to specific categories. Therefore, incorporating CWE entries allows the model to directly learn vulnerability-related knowledge. In the dataset, the CWE categories corresponding to NSVRs are saved to construct anchors. Each anchor has at least one NSVR associated with it. For each CWE entry, we extract relevant attributes as part of the vulnerability knowledge base. Each anchor includes the CWE category and its corresponding weakness attributes. The collection of all anchors forms the Knowledge Bank, which provides vulnerability information to the model.

We utilize the following five attributes to construct anchors: Name refers to the name of the weakness. The description refers to how the weakness arises. Extended Description refers to the additional description of the weakness. Common Consequences refer to negative impacts caused by the weakness. Related Weaknesses refer to relationships between the weakness and other weaknesses.

### C. Matching component

The Matching component compares the input IR with different CWE categories and outputs a scoring distribution measuring similarity between IR descriptions and anchors. We use a Siamese network to directly compare the similarity between two information items or sentences, with IR and anchor content as inputs.

The Siamese network architecture consists of two parts: feature extraction and matching. The feature extraction part

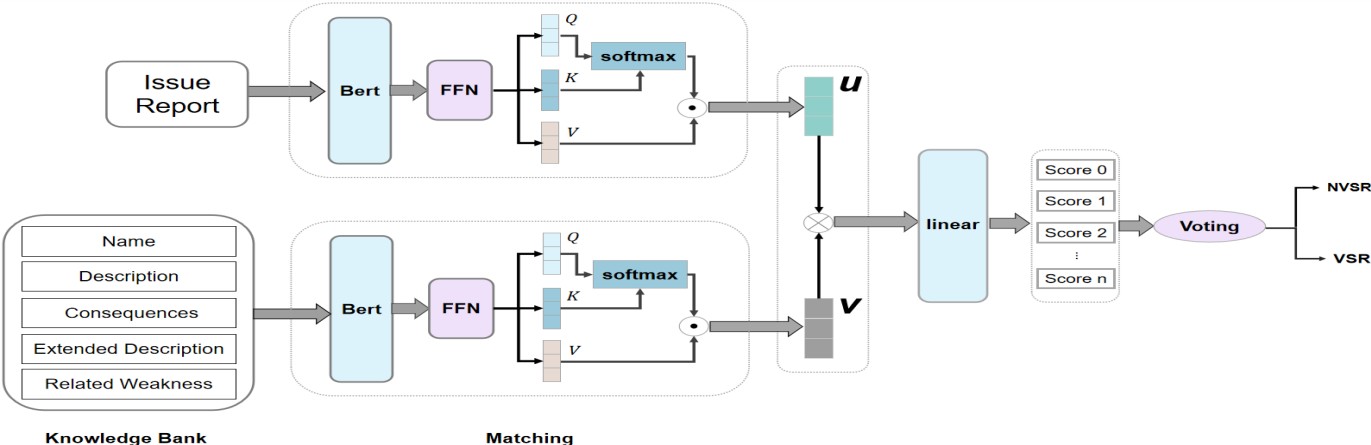

Fig. 2. Framework of Vuln-Detector.

extracts feature vectors from the input IR and anchor content. The matching part calculates the similarity probability between IR and anchor content based on the extracted vectors. Both inputs share the same network weights at the encoding layer, transforming them into the same feature space.

Positive samples account for less than 1% of the dataset, leading to an imbalanced sample distribution. We introduce a metric-based few-shot learning technique to effectively utilize the limited number of NSVRs and improve recognition accuracy.

*1) Feature Extraction:* Our goal is to match the input IR with different CWE categories. During training, both IR and CWE information are input into the model for classification. Since IRs and anchor content are natural language descriptions, we use BERT to convert them into feature vectors. BERT, a pre-trained model based on the transformer's encoder, effectively extracts feature vectors from sentences for classification tasks [15]. It learns language representations from large-scale unlabeled text data and transfers this knowledge to downstream tasks [16]. We fine-tune BERT on 1,221,677 IRs to identify vulnerable reports. During training, BERT's attention mechanism parameters are updated. BERT uses sentences as inputs, incorporating the [CLS] token at the beginning and the [SEP] token at the end of each sentence.

*2) Classification Matching :* In this component, we aim to measure the matching degree between the IR and the anchor information of different categories. Previous approaches for matching IR and anchors often use Feed-Forward Networks (FFNs) as classifiers [7], [12]. In our model, the feature vectors obtained from BERT encoding are first input into a fully connected feed-forward module and then into a self-attention module. Subsequently, we input the resulting feature vectors of the vulnerability report text into a fully connected layer for further processing, followed by a softmax classifier. The cross-entropy loss function is used in the classification task.

To enhance the model's fitting ability, we implemented several procedures. Feedforward neural networks capture higher-level correlations by mapping input to output through com-

positions of nonlinear functions. We applied a feedforward neural network (with shared parameters) to the feature vectors of IR and anchor information obtained from BERT's output to introduce nonlinearity and consider interactions between latent dimensions.

We incorporated an attention mechanism into the classifier, helping the model dynamically focus on specific parts of the input sentence in text classification tasks [17]. This mechanism assigns higher weights to relevant information, improving focus on important data while suppressing irrelevant details.

To address long-distance dependencies and capture word relevance in IRs, we introduce a self-attention mechanism. Self-attention explores hidden correlations within feature data and can directly link relationships between any two words in a single computational step, enhancing the model's ability to attend to different positions in the text [18].

To further obtain the relevance between words in the vulnerability information text, we input the feature vectors obtained from the FFN layer of the IR and anchor them into the self-attention mechanism. Let's assume the input feature vector is $X = [x_1, x_2...x_N] \in R^{D_x} \times N$, which is copied three times and multiplied with $W^Q, W^K$, resulting in $Q, K, V$,

$$Q = W^Q X \in R^{D_Q \times N} \tag{1}$$

$$K = W^K X \in R^{D_K \times N} \tag{2}$$

$$V = W^V X \in R^{D_V \times N} \tag{3}$$

Then, the obtained $Q, K, V$ are input into the attention layer. The attention layer first calculates the dot product of $Q$ and $K$, and then divides the product by $\sqrt{d_k}$, to prevent the dot product from becoming too large as the vector length increases. Subsequently, the softmax function is applied to the resulting values to obtain a weight distribution with a sum of 1. Finally, the obtained weight is multiplied to obtain the relevance between words in the vulnerability information text. This enables the model to focus on features with higher

weights while suppressing features with lower weights, and it can be expressed as

$$Attention(Q, K, V) = softmax(\frac{QK^T}{\sqrt{d_k}})V \qquad (4)$$

where $Q, K, V$ are the input vulnerability feature vectors, $W^Q, W^K, W^V$ are the results of element-wise multiplication between, and $d_k$ is the first dimension of matrix $W$.

To utilize different feature information from various positions in the vulnerability data, we introduce a multi-head self-attention mechanism. After self-attention, we extract feature vectors of the IR and anchor content, improving the capture of relationships between distant sequence elements and enhancing classification accuracy. We designate the input IR feature vector as $u$ and the anchor content feature vector as $v$. The joint feature vector $(u, v, |u - v|)$ is created by concatenating these vectors and is denoted as $x$. This vector is then fed into the model's top layer via a linear layer and passed through the softmax function. The softmax operation converts the input joint vector into probability distributions, reflecting the likelihood of reported IR matches with different categories of CWEs. This process is expressed as

$$softmax(x_i) = \frac{exp(x_i)}{\sum\limits_{i=1}^{M} exp(x_i)} \qquad (5)$$

where $x_i$ is the value of the $i$th input node of the feature vector $x$, and $M$ is the total number of output nodes, i.e., the number of NSVR species.

### D. Voting component

The Voting component determines the category (NSVR or SVR) of the report's IR. We input the matching scores obtained from the classifier into the Voting component and train a threshold. Assuming we have NSVR categories, the process is

$$f(x) = \underset{c \in C}{\operatorname{argmax}} \sum_{i=1}^{n} p_{i,c} \qquad (6)$$

where $C$ represents the set of output NSVR categories, and $p_{i,c}$ represents the probability of outputting a certain NSVR category.

After the aforementioned processing, we compare the highest obtained matching score with the threshold. If the matching score is higher than the threshold, the report's IR is classified as NSVR. If no matching score reaches the threshold, indicating that there is no candidate vulnerability type for matching, the IR is predicted as SVR.

### IV. EXPERIMENT SETUP

#### A. Research Questions

In our experiments, we aimed to answer the following questions:

**RQ1:** Is our proposed deep learning-based approach effective in identifying NSVRs?

**RQ2:** What is the impact of the pooling method in BERT on the model's performance?

**RQ3:** How many layers of self-attention are most effective?

**RQ4:** What is the optimal number of heads in the multi-head self-attention mechanism?

**RQ5:** How does the impact of the self-attention mechanism on the model compare to that of other components?

#### B. Data Preparation

We collected CWE records from the National Vulnerability Database (NVD) and filtered out those not referenced in GitHub IRs, ensuring data validity by excluding IRs with missing information and NSVRs created after their corresponding CVE disclosure dates. This resulted in a final dataset of 1,195,202 vulnerability reports from 1,360 projects, divided into training, validation, and testing sets. As shown in Table I, we divided the dataset into three sets, for training, validation, and testing.

TABLE I
DATA SET PARTITIONING

|  | NSVR | SVR | Projects |
|---|---|---|---|
| **Training Set** | 3,175 | 969,570 | 1,102 |
| **Validation Set** | 306 | 103,273 | 122 |
| **Testing Set** | 403 | 118,475 | 136 |

#### C. Baseline

To evaluate the effectiveness of our proposed approach (RQ1), we compared it against the following baseline approaches:

(1) TextCNN [19]: TextCNN has gained widespread use in software engineering and it has been shown to achieve good results in text classification so we use it as a neural network baseline. This approach employs multiple convolutional filters of varying sizes to capture N-gram information from the text. It utilizes max-pooling to extract the most salient features from each convolutional operation. These features are then concatenated and fed into a fully connected layer for the feature. The model is trained using cross-entropy loss.

(2) Random Guess [20]: Stochastic prediction for demonstrating the effectiveness of deep learning approaches. This baseline randomly predicts whether the input IR belongs to the NSVR class.

(3) Naive Bayes (NB) [21]: It is shown that simple text categorization methods outperform methods designed in previous studies on clean datasets [22]. This approach is a simple text classification technique based on Bayes' theorem.

(4) MemVul [12]: A more advanced approach based on deep learning proposed by Pan et al. It is shown to be the optimal model for identifying vulnerability reports obtained from a combination of metrics.

To investigate the impact of different pooling strategies on model performance (RQ2), we explored four pooling methods:

(1) CLS token pooling: The vector representation of the [CLS] token is used as the sentence vector.

(2) Average pooling: The average of all token vectors is computed to obtain the sentence vector.

(3) Attention pooling: Attention weights are learned to dynamically weight the token vectors and compute the sentence vector.

(4) Max pooling: The maximum value among all token vectors is selected as the sentence vector.

### D. Evaluation Metrics

We employed five evaluation metrics, namely Precision, Recall, F1-score, AUROC, and AUPRC, to assess our results. Among them, Precision, Recall, and F1-score are commonly used metrics in software engineering, including vulnerability report classification [23]. Precision measures the accuracy of positive predictions, while Recall evaluates the proportion of correctly predicted positive samples out of all actual positive samples. The expressions of Precision and Recall are

$$Precision = \frac{TP}{TP + FP} \tag{7}$$

$$Recall = \frac{TP}{TP + FN} \tag{8}$$

In the formulas, $TP$ represents true positives (i.e., correctly predicted positive samples), $FP$ represents false positives (i.e., incorrectly predicted positive samples), and $FN$ represents false negatives (i.e., incorrectly predicted negative samples).

We utilized the F1-score as a comprehensive evaluation metric. The F1-score represents the model's ability to strike a balance between Precision and Recall, with higher values indicating a stronger ability to improve both metrics simultaneously. Previous work [22] has also highlighted the F1-score as an important metric for evaluating vulnerability report classification.

$$F1 - score = \frac{2 \times precision \times recall}{precision + recall} \tag{9}$$

AUPRC measures the model's ability to balance Precision and Recall. AUROC is a metric used to assess classifier performance, with higher values indicating a stronger ability to correctly discriminate between positive and negative samples.

### E. Training Details

We performed MLM (Masked Language Model) tasks on the obtained IRs for 50 epochs and employed AdamW [24] as the optimizer. The learning rate (lr) for BERT was set to $2e^{-5}$. The model consisted of two layers of self-attention, with each layer having a head count of 8. The FFN (Feed-Forward Network) dimension was set to 2048. Additionally, following the work of Neculoiu et al. [25], we trained the Siamese network with the ratio of non-matching to matching pairs by 3:1.

## V. EXPERIMENT RESULTS

### A. RQ1: Is our proposed deep learning-based approach effective in identifying NSVRs?

**Motivation** The purpose of our proposed approach is to automatically identify NSVRs at an early stage to prevent the leakage of vulnerability information, which may cause

harm to OSS users. We aim to validate the effectiveness of Vuln-Detector and demonstrate that our approach outperforms previous relevant approaches.

**Approach** We collect 1,221,677 vulnerability information from NVD, CWE, and GitHub IRs related to the Vuln-Detector OSS project. To test the approach's identification performance, we input IRs from the Vuln-Detector project, including 403 NSVRs and 118,475 SVRs. TextCNN, Random Guess (RG), Naive Bayes (NB), and MemVul are used as baselines. To make a fair comparison, we use the same dataset and evaluation metrics to replicate the baselines' results. We evaluate the approach's performance using five metrics: Precision, Recall, F1-score, AUROC, and AUPRC.

**Results** The comparison between our approach and the baselines in terms of various performance metrics is shown in Table II. For ease of comparison, we bolded the best result for each metric.

TABLE II
RESULTS OF VULN-DETECTOR AND COMPARISON WITH BASELINE

| Approach | Precision | Recall | F1-score | AUROC | AUPRC |
|---|---|---|---|---|---|
| MemVul | 0.38 | 0.70 | 0.49 | 0.98 | 0.39 |
| TextCNN | 0.18 | 0.73 | 0.28 | 0.97 | 0.27 |
| RG | 0.003 | 0.50 | 0.01 | 0.50 | 0.003 |
| NB | 0.16 | **0.81** | 0.26 | 0.95 | 0.23 |
| Ours | **0.42** | 0.73 | **0.53** | **0.98** | **0.41** |

The experimental results, as shown in Table II, demonstrate that Vuln-Detector outperforms the baselines in terms of Precision, F1-score, AUROC, and AUPRC. Vuln-Detector achieves a Precision of 42%, which is an improvement of 11% over the best baseline result. This indicates that our approach can more accurately identify NSVRs. Our approach achieves an F1-score of 53%, an improvement of 8% over the best baseline result. The AUPRC is 41%, which is 5% higher than the best baseline result. The improved F1-score and AUPRC results indicate that our approach strikes a good balance between Precision and Recall, ensuring that the model can identify more NSVRs while accurately identifying them.

In terms of baseline performance, it is observed that RG performs poorly across all five metrics. This can be attributed to its reliance solely on random prediction for IR classification, lacking a solid theoretical foundation. On the other hand, NB achieves the highest recall rate of 81% among the five approaches, but exhibits poor Precision, F1-score, and AUPRC. This approach relies on probability theory as its theoretical basis and only utilizes simple IR features for classification, without considering the correlations between individual features. While it may effectively avoid missing NSVR, its recognition accuracy is significantly inadequate, failing to strike a good balance between accuracy and recall. The TextCNN model has a simpler structure, but struggles with the processing of long texts, resulting in the loss of important features and the inability to capture long-distance dependencies. Consequently, it achieves a high recall rate but only 18% precision. In contrast, MemVul demonstrates superior performance across all metrics by leveraging the BERT language

model to extract features from vulnerability reports. BERT exhibits robust linguistic characterization and feature extraction capabilities, effectively capturing the semantic information present in vulnerability reports. This highlights the significance of the feature extraction component in the vulnerability report identification model. However, when compared with Vuln-Detector, MemVul obtains poorer results in various metrics. This discrepancy arises from the incorporation of the self-attentive mechanism feature extraction component into our model. This additional component performs post-processing on the features, extracting more detailed IR characteristics and enhancing the model's nonlinear capacity to capture global contextual information. As a result, MemVul acquires richer semantic information, thereby facilitating improved vulnerability report classification.

**Conclusion** Vuln-Detector demonstrates effectiveness in predicting the risk level of incoming vulnerability reports, thereby reducing the likelihood of open source software being targeted by attackers. Moreover, when compared to the baselines, our model achieves superior results across various metrics. This further validates the efficacy and practicality of our proposed approach over existing approaches.

*B. RQ2: What is the impact of the pooling method in BERT on the model's performance?*

**Motivation** The pooling method in BERT can have an impact on the overall performance. Pooling is a feature selection and information filtering process, and different pooling methods can yield different computational speeds and experimental results. Therefore, it is necessary to explore different pooling methods in Vuln-Detector to identify the optimal one.

**Approach** In the experiment, we employed four different pooling methods. The first method is mean pooling, which calculates the average of the output vectors of all tokens to represent the sentence vector. The second method is max pooling, which selects the maximum value for each dimension of the output vectors as the sentence vector. The third method is attention pooling, which selectively aggregates values to generate the output. The fourth method directly uses the output vector at the CLS position to represent the entire sentence. Except for the different pooling methods, all other module settings and training details remain the same. We evaluated the best pooling method based on five evaluation metrics: Precision, Recall, F1-score, AUROC, and AUPRC.

**Results** The results of different pooling methods are shown in Table III. For ease of comparison, we bolded the best result for each metric.

The experimental results, shown in Table III, indicate that different pooling methods affect the model's results. The CLS pooling method outperforms the other three pooling methods in terms of Recall, F1-score, and AUROC. Specifically, the CLS pooling method achieves a Recall of 73%, which is an improvement of 18% compared to the best results obtained by the other three pooling methods. A higher Recall enables the model to identify more NSVRs, reducing the probability of missing critical vulnerability reports. Our method achieves an

TABLE III
COMPARISON OF DIFFERENT POOLING METHODS

| POOLING | Precision | Recall | F1-score | AUROC | AUPRC |
|---|---|---|---|---|---|
| Mean | 0.46 | 0.62 | 0.53 | 0.97 | 0.41 |
| Max | **0.48** | 0.58 | 0.53 | 0.97 | 0.41 |
| Attention | 0.45 | 0.61 | 0.52 | 0.97 | 0.41 |
| CLS | 0.42 | **0.73** | **0.53** | **0.98** | **0.41** |

F1-score of 53%, indicating a good balance between Precision and Recall using our pooling method. The AUROC is 98%, indicating the strong capability of our model to correctly identify both risky and safe vulnerability reports. The AUPRC values for all four pooling methods are 41%.

Max pooling has the highest Precision but the lowest Recall among the four pooling methods. Max pooling selects the maximum value for each dimension, increasing the receptive field and extracting more useful feature information, resulting in the highest identification accuracy. Precision and Recall are conflicting metrics, so a higher Precision also implies lower Recall. Mean pooling and Attention pooling have lower Recall rates, as both methods may blur the textual features of the input vulnerability reports, causing the model to miss some NSVRs due to insufficient linguistic information. CLS pooling yields the lowest Precision, possibly due to the direct use of the output vector at the CLS position to represent the entire sentence, leading to the loss of some features. However, CLS pooling achieves the highest values for the other four metrics. This is because, compared to other pooling methods, CLS pooling is more effective at capturing contextual information under specific contexts, resulting in higher Recall and F1-score values, meaning it can identify more risky vulnerability reports and reduce the risk of losses for OSS users due to missed critical vulnerability reports.

**Conclusion** In this experiment, the CLS pooling method performed the best in Vuln-Detector compared to other pooling methods.

*C. RQ3: How many layers of self-attention are most effective?*

**Motivation** The self-attention mechanism is a key factor in improving the performance of Vuln-Detector. The number of attention layers can influence the model's learning capacity for semantic features. To maximize the performance of the model, we aim to understand the impact of the number of attention layers.

**Approach** In this experiment, we vary the number of self-attention layers to 1, 2, and 3, respectively. Apart from the different number of self-attention layers, all other module settings and training details remain the same. In this experiment, we evaluate the model using the five evaluation metrics: Precision, Recall, F1-score, AUROC, and AUPRC.

**Results** The results of different self-attention layers are shown in Table IV. For ease of comparison, we bolded the best result for each metric.

The experimental results, as shown in Table IV, reveal that the model's performance fluctuates with the variation in the

TABLE IV
COMPARISON OF DIFFERENT SELF-ATTENTION LAYERS

| NUM | Precision | Recall | F1-score | AUROC | AUPRC |
|-----|-----------|--------|----------|-------|-------|
| 1 | **0.44** | 0.70 | **0.54** | 0.98 | **0.46** |
| 3 | 0.40 | 0.55 | 0.46 | 0.98 | 0.36 |
| 2 | 0.42 | **0.73** | 0.53 | **0.98** | 0.41 |

TABLE V
COMPARISON OF RESULTS WITH DIFFERENT NUMBERS OF HEADS

| HEAD NUM | Precision | Recall | F1-score | AUROC | AUPRC |
|----------|-----------|--------|----------|-------|-------|
| **8head** | 0.44 | 0.69 | **0.54** | 0.98 | **0.43** |
| **16head** | **0.47** | 0.60 | 0.53 | 0.98 | 0.43 |
| **4head** | 0.42 | **0.73** | 0.53 | **0.98** | 0.41 |

number of self-attention layers. We find that when the number of self-attention layers is 2, the model performs well across all five metrics. Specifically, it achieves a Recall of 73%, which is 4% higher than the best results obtained by other layer numbers. This implies that the model can identify more risky vulnerability reports when there are 2 self-attention layers. The AUROC is 98%, indicating the strong ability of the model to distinguish between SVRs and NSVRs.

When the number of layers is 1, the Precision, F1-score, and AUPRC are better than the results obtained by other layer numbers. However, the Recall is only 70%, which does not show improvement compared to the best baseline result. This is because with only one layer, the network depth is shallow, and the crucial features of vulnerability reports may be misled by other features, resulting in lower Recall. When the number of layers is 3, the Precision is 40%, Recall is only 55%, AUPRC is 36%, and F1-score is 46%. These results are the worst among the three different layer numbers. This is because an excessive number of layers enhances the features of input vulnerability reports, but it also means that the model requires more effective features from the reports to judge risky vulnerability reports. This may lead to the model failing to recognize some risky vulnerability reports that lack sufficient relevant features. Additionally, with a small number of positive samples in the dataset, having too many layers may amplify irrelevant features of negative samples, resulting in decreased model performance.

**Conclusion** When there are 2 self-attention layers, the model achieves the highest Recall and performs well in other metrics. It strikes a good balance between layer numbers 1 and 3. Therefore, we ultimately set the number of self-attention layers to 2 in our Vuln-Detector model.

### D. RQ4: What is the optimal number of heads in the multi-head self-attention mechanism?

**Motivation** The self-attention mechanism is an important component that affects the performance of the Vuln-Detector model. The number of heads in each layer of self-attention can impact the model's ability to learn different positional and semantic features. In order to maximize the model's performance, we aim to investigate the effect of the number of heads and identify the optimal value.

**Approach** In this experiment, we set the number of heads to 4, 8, and 16, respectively. Apart from the different number of heads, all other module settings and training details remain the same. In this experiment, we still use the evaluation metrics of Precision, Recall, F1-score, AUROC, and AUPRC to assess the model.

**Results** The results obtained with different numbers of heads are presented in Table V. To facilitate comparison, we highlight the best result for each metric in bold.

The experimental results are shown in Table V. Firstly, it can be observed from the table that the model performance varies with the number of heads. We found that when the number of heads is 4, the model achieves good results for all five metrics. Specifically, the Recall value is 73%, which is 6% higher than the best result obtained with other numbers of heads. This indicates that the model can identify more dangerous vulnerability reports when the number of heads is 4, making it more suitable for real-world scenarios. The AUROC is 98%, demonstrating the model's strong ability to distinguish between NSVR and SVR.

Compared to the case with 4 heads, the highest Precision is achieved when the number of heads is 16, but its Recall value is only 60%, indicating the poorest performance. When the number of heads is 8, the F1-score and AUPRC values perform the best, but the Recall is still lower compared to the case with 4 heads. It can be observed that as the number of heads increases, the Recall decreases. This is because the enhancement of the number of heads leads to the extraction of more irrelevant features, thereby overlooking many crucial features.

**Conclusion** In conclusion, compared to other numbers of heads, the model performs best when the number of heads is 4. Therefore, we set the number of heads to 4 in each layer of the self-attention mechanism.

### E. RQ5:How does the impact of the self-attention mechanism on the model compare to that of other components?

**Motivation** The self-attention mechanism in Vuln-Detector is our key design. To compare it with other feature extraction components and further demonstrate the effectiveness of our key design, we replace the self-attention feature extraction component with three other feature extraction approaches. In this experiment, we still use five evaluation metrics: Precision, Recall, F1-score, AUROC, and AUPRC to evaluate the model.

**Approach** In this experiment, we set the feature extraction components to be BiLSTM, FFN, TextCNN, and self-attention, respectively. Apart from the different feature extraction components, the settings and training details of other modules remain the same.

**Results** The results obtained with different feature extraction components are shown in Table VI. To facilitate comparison, we highlight the best result for each metric using bold and underline.

The experimental results are shown in Table 7. Self-attention outperforms other feature components in terms of F1-score

TABLE VI
PERFORMANCE WITH DIFFERENT FEATURE EXTRACTION COMPONENTS.

| DIFF MODEL | Precision | Recall | F1-score | AUROC | AUPRC |
|---|---|---|---|---|---|
| BiLSTM | 0.37 | 0.74 | 0.50 | 0.96 | 0.27 |
| FFN | **0.44** | 0.62 | 0.51 | 0.97 | **0.43** |
| TextCNN | 0.30 | **0.79** | 0.44 | 0.98 | 0.37 |
| Self-attention | 0.42 | 0.73 | **0.53** | **0.98** | 0.41 |

and AUPRC. Specifically, the F1-score is 53%, which is 4% higher than the best result obtained by other components. This indicates that our key design improves the model's ability to balance between Precision and Recall. Furthermore, the AUPRC is 98%, further demonstrating the strong balance between Precision and Recall in our approach.

In comparison to Self-attention, BiLSTM [26] performs inadequately across all metrics. Despite being an improvement over traditional RNNs, BiLSTM still struggles to effectively transmit information from the start of excessively long sequences. On the other hand, FFN achieves the highest Precision and AUPRC values, but its Recall value is only 62%, indicating the poorest performance among the considered feature components. This drawback can be attributed to the simplified network structure of FFN, which is not well-suited for extracting feature vectors from lengthy texts, causing the loss of some semantic information and resulting in a low recall rate.

TextCNN attains the highest Recall value, but its Precision value is only 30%, the lowest among the four feature extraction components. This suggests its limited accuracy in correctly predicting dangerous vulnerability reports. In contrast, the Self-Attention mechanism exhibits the best overall performance when compared to the other three feature extraction components. This superiority can be attributed to the self-attention mechanism's ability to calculate similarity without dependencies, enabling efficient parallel computation and significantly improving computational efficiency. Furthermore, the introduction of the self-attention mechanism addresses the challenge of capturing long-distance dependencies in texts, thereby mitigating the loss of semantic information. Consequently, it facilitates the effective extraction of feature vectors from input vulnerability reports.

**Conclusion** Self-attention is the most effective feature extraction component to improve the model's expressive ability to make the best model performance.

## VI. THREATS TO VALIDITY

In this section, we analyze the threats to validity from three aspects: internal validity, external validity, and construct validity.

### A. Internal Validity

The main threat to internal validity is the correctness of Vuln-Detector and the replication of the model by Pan [12]. We conducted multiple code inspections to minimize errors in Vuln-Detector. Additionally, we replicated Pan's approach, and although the results may differ from the original paper

due to device or other factors, we still compared our results with the original paper, which may introduce some impact.

### B. External Threats

The main threat to external validity is that we only experimented with dangerous IRs from the GitHub IRs referenced by CVEs. There are many other vulnerability information databases besides CVE, so our dataset may not represent all dangerous IRs, and we cannot be certain about the generalizability of our approach.

However, we believe in the effectiveness of Vuln-Detector in early automatic identification of dangerous vulnerability reports. In the future, we plan to collect more datasets from different open-source platforms and evaluate Vuln-Detector in real-world development scenarios to address this limitation.

### C. Construct validity

The main threat to construct validity is the lack of handling the imbalance between positive and negative samples in Vuln-Detector. The scarcity of dangerous vulnerability reports may affect the approach's ability to correctly identify them. In future work, we plan to take measures to increase the number of dangerous vulnerability reports.

## VII. RELATED WORK

In this section, we delve into the automatic mining and classification of vulnerability reports and comments in online chatrooms, two crucial areas in open-source software (OSS) project development. Section A discusses the significance of accurately identifying and classifying vulnerability reports, reviewing various approaches and techniques used in previous research. Section B turns to the automatic classification of comments in online chatrooms, a common communication channel among OSS developers.

### A. Automatic Mining of Vulnerability Reports

Previous research has highlighted the importance of automatically identifying dangerous vulnerability reports and proposed approaches for classifying security and non-security bug reports. Most existing work is based on bug repositories, treating bug classification as text classification. Kudjo et al. [27] demonstrated the challenges with TF-IDF and proposed a vulnerability classification approach based on term frequency and inverse gravity moment (TF-IGM) to improve the classification accuracy of the model. Patrick et al. [28] proposed the use of a security vocabulary specific to a particular project to enhance the importance of standard security keywords. Subsequently, Oyetoyan et al. [29] demonstrated that training text classification models with security keywords can improve the generalization ability of the model.

Based on the discussions of the above previous work, it can be observed that most of the previous approaches focus on optimizing the feature extraction component. This indicates that the quality of extracted features plays a significant role in the classification results of vulnerability reports. The problem of how to extract, retain, and enhance effective features remains an area for further exploration.

Compared to previous work, we adopt a Siamese network model structure, utilize BERT encoding, and input the encoded feature vectors into the self-attention mechanism to integrate and enhance the IR feature information. Additionally, our approach can capture global information from the input IR sequences, addressing the issue of word correlations and improving the accuracy of the model.

### B. Automatic Classification of Comments in Online Chatrooms

Online chatrooms serve as communication channels among most OSS project developers. Platforms such as Slack [30] and Gitter [31] are commonly used for communication. Lin et al. [32] explored how Slack influences the dynamics of development teams and conducted an exploratory study to understand how developers use Slack and benefit from it. They found that Slack plays a significant role in software development, enabling developers to communicate OSS project information through instant messaging and discuss and resolve bug reports. Similar to bug reports, the chat messages in online chatrooms are often diverse, and classifying them for easier access to desired information becomes necessary.

Antoniol et al. [33] used Alternating Decision Trees (an extension of decision trees), Naive Bayes classifiers, and logistic regression to build a classifier that categorizes reports into bug and non-bug reports. Gu et al. [34] proposed SUR-Miner, a model that goes beyond the assumption of a simple bag-of-words model and categorizes chat message data into five different classes: aspect evaluation, praise, feature request, error report, and other. Arya et al. [35] employed TF-IDF weighting as text features and introduced several machine learning algorithms to classify sentences in problem statements. Shi et al. [36] proposed a Siamese network-based approach to transform the traditional text classification task of mapping individual dialogs to their categories into a task of determining whether two dialogs are similar by learning from a few-shot merging. Sorbo et al. [37] introduced intent mining, and based on that, Huang et al. [38] improved it by proposing a CNN-based approach for content classification of comments. This approach achieved a 171% improvement in accuracy compared to the approach proposed by Sorbo et al. [37].

Compared to the aforementioned studies, our approach focuses on categorizing vulnerability reports into dangerous and non-dangerous reports, providing more precise classification results than the studies mentioned above. We pay closer attention to the structural and content information of the reports, categorizing them not only as bug reports but also explicitly classifying them as either dangerous or non-dangerous vulnerability reports.

## VIII. CONCLUSION AND FUTURE WORK

The disclosure of dangerous vulnerabilities on the internet before being remedied can lead to attackers exploiting the vulnerability information, resulting in losses for OSS users. To automatically identify dangerous vulnerability reports in the early stages, we proposed an approach called Vuln-Detector, which effectively recognizes dangerous vulnerability reports. The approach consists of a feature extraction component and a classification component. The feature extraction component is responsible for extracting feature vectors from the input vulnerability report texts, and we incorporated self-attention mechanisms to enhance the model's ability to extract textual features. The classification component matches the input vulnerability report with known dangerous vulnerabilities, and if there is no corresponding known dangerous vulnerability report, it is classified as a safe report; otherwise, it is classified as a dangerous vulnerability report. Experimental results based on vulnerability reports from 1,360 projects on GitHub showed that Vuln-Detector outperformed the approach by Pan et al. [12] in terms of Precision, F1-score, AUROC, and AUPRC, with improvements of 11%, 8%, and 5%, respectively. Qualitative analysis results also demonstrated that Vuln-Detector performed better than the baselines in identifying dangerous vulnerability reports.

For future work, we plan to propose better data augmentation methods to address the severe class imbalance issue in the dataset and train more effective models. Additionally, we will use datasets from different vulnerability information platforms to train our model so that it can be effectively applied to various OSS platforms.

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
