# OpenReview forum: "Identifying Meaningful Vulnerability Report in Common Weakness Enumeration"
_IEEE.org/ICIST/2024/Conference — IEEE ICIST 2024 Conference Submission_

### Official Review · Reviewer_DFfq · 2024-08-22
**This article is very interesting and a good one**

**Rating:** 7
**Confidence:** 3

**Review:**

This study introduces Vuln-Detector, a pioneering approach that automates the identification of hazardous IRs. The results conclusively show that Vuln-Detector proficiently detects vulnerability-associated IRs, underscoring its effectiveness and potential impact. The obtained result is valuable and can be accepted if the following problems can be clarified.
(1) In the introduction, it is not enough to state the current work. It should be expended and reconstructed.
(2) The authors are suggested to add a research framework diagram to facilitate readers' reading
(3) There are some typos and grammar errors. The authors should have a native English speaker or software packages to perform the editing check.
(4) The references need to be updated and the format needs to be standardized.

---

### Official Review · Reviewer_qQe3 · 2024-08-22
**Due to previous models often used shallow neural networks with a single feature extraction method, failing to capture deep feature representations. To address this, this paper proposed Vuln-Detector, an approach for automatically identifying dangerous Issue Reports (IRs). The topic of this paper is interesting.**

**Rating:** 8
**Confidence:** 3

**Review:**

Comments to the Author
Due to previous models often used shallow neural networks with a single feature extraction method, failing to capture deep feature representations. To address this, this paper proposed Vuln-Detector, an approach for automatically identifying dangerous Issue Reports (IRs). The topic of this paper is interesting. Below is a list of comments that should be taken into account further when revising the paper.
1.	The contribution of this article should be compared with previous literature, and the basic technical difficulties of this article should be listed? And what methods should be used to solve this problem, emphasizing novelty and technological contribution.
2.	This article introduces a self-attention mechanism, which should be elaborated to enable readers to understand some of its related performance.
3.	In the conclusion section, all mechanisms should be compared. Finally, self-attention is the most effective feature extraction component to improve the model’s expressive ability to make the best model performance.

---

### Official Review · Reviewer_FL7P · 2024-08-23
**Identifying Meaningful Vulnerability Report in Common Weakness Enumeration**

**Rating:** 7
**Confidence:** 2

**Review:**

To  address the vulnerabilities in Open Source Software code，this paper proposed an approach for automatically identifying dangerous Issue Reports. The approach consists of a feature extraction component and a classification component. There are some problems that should be replied. Comments for this submission are given as follows:
1. The paper should include comparisons against the existing literature to demonstrate its advantages.
2.To investigate the impact of different pooling strategies on model performance, the paper explored four pooling methods. Can you explain why these four types of pooling methods were chosen?
3. The architecture of Vuln-Detector consists of three components: a) Knowledge Bank, b) Matching component, and c) Voting component. Please explain in detail the connection between these three parts.

---

### Comment · Reviewer_DFfq · 2024-08-21
**This article is very interesting and a good one**

This study introduces Vuln-Detector, a pioneering approach that automates the identification of hazardous IRs. The results conclusively show that Vuln-Detector proficiently detects vulnerability-associated IRs, underscoring its effectiveness and potential impact. The obtained result is valuable and can be accepted if the following problems can be clarified.
(1)	In the introduction, it is not enough to state the current work. It should be expended and reconstructed.
(2)	The authors are suggested to add a research framework diagram to facilitate readers' reading
(3)	There are some typos and grammar errors. The authors should have a native English speaker or software packages to perform the editing check.
(4)	The references need to be updated and the format needs to be standardized.

---

### Decision · Program_Chairs · 2024-09-06

Accept (Oral)